# An Overview of Modified Chitosan Adsorbents for the Removal of Precious Metals Species from Aqueous Media

**DOI:** 10.3390/molecules27030978

**Published:** 2022-02-01

**Authors:** Dexu Kong, Stephen R. Foley, Lee D. Wilson

**Affiliations:** 1Saskatchewan Research Council, 125-15 Innovation Boulevard, Saskatoon, SK S7N 2X8, Canada; dek593@mail.usask.ca; 2Department of Chemistry, University of Saskatchewan, 110 Science Place, Saskatoon, SK S7N 5C9, Canada

**Keywords:** modified adsorbents, hybrid inorganic materials, chitosan, complex formation, selectivity, gold complexes, adsorption

## Abstract

This mini-review provides coverage of chitosan-based adsorbents and their modified forms as sustainable solid-phase extraction (SPE) materials for precious metal ions, such as gold species, and their complexes in aqueous media. Modified forms of chitosan-based adsorbents range from surface-functionalized systems to biomaterial composites that contain inorganic or other nanomaterial components. An overview of the SPE conditions such as pH, temperature, contact time, and adsorbent dosage was carried out to outline how these factors affect the efficiency of the sorption process, with an emphasis on gold species. This review provides insight into the structure-property relationships for chitinaceous adsorbents and their metal-ion removal mechanism in aqueous media. Cross-linked chitosan sorbents showed a maximum for Au(III) uptake capacity (600 mg/g), while S-containing cross-linked chitosan display favourable selectivity and uptake capacity with Au(III) species. Compared to industrial adsorbents such as activated carbon, modified chitosan sorbents display favourable uptake of Au(III) species, especially in aqueous media at low pH. In turn, this contribution is intended to catalyze further research directed at the rational design of tailored SPE materials that employ biopolymer scaffolds to yield improved uptake properties of precious metal species in aqueous systems. The controlled removal of gold and precious metal species from aqueous media is highly relevant to sustainable industrial processes and environmental remediation.

## 1. Introduction

Precious metals such as gold (Au), silver (Ag), platinum (Pt), and palladium (Pd) are highly valued by industry, medicine, coinage, and jewelry because of their specific physical and chemical properties. In the electronics industry, solid-state devices often use very low currents and voltage, where the slightest corrosion can occur at the contact points, which can disrupt the current flow. Gold is known to resist corrosion; its utility as a reliable conductor in producing high-end electronics (smartphones, speakers, etc.) and specialized connectors is well established. Currently, gold is globally recognized as a form of currency metal, where the role of precious metal production and recycling from primary or secondary sources have a tremendous impact on the environment and the global economy.

The limited availability of these precious metals from natural resources makes their recovery valuable and significant. To improve the utilization of metals and avoid unnecessary mining from primary sources, recent efforts are directed at the recovery of metals from secondary sources: low-grade ores, electronic waste (e-waste), computer scrap, mine tailings, and industrial wastes. In turn, current research is directed at the development of technology for the recovery of precious metals from wastewater, including pyrometallurgical and hydrometallurgical processes [1]. Hydrometallurgical methods are widely applied by industry because of their low capital cost, low toxic gas emissions, and facile unit operations. In the case of the hydrometallurgical methods, there are three key steps in the process: (1) leaching of precious metals; (2) scrubbing to remove unwanted metals; and (3) stripping to remove the extracted target metal species from the organic phase. Upon the transfer of the precious metals to the liquid phase, hydrometallurgical processing can be applied directly to the solution media. In many cases, precious metals are present in their solid state, such as ores or scrap metals. To afford the dissolution of gold ions, leaching with common agents such as aqua regia, hydrogen peroxide, cyanide, halides, thiourea, and thiosulfate may be employed. Cyanide leaching (cf. Table 6 in [1]) is an effective way to dissolve precious metals, but due to its high toxicity, contamination of water resources may occur upon its uncontrolled release into the environment, which can negatively affect aquatic biota [1]. An alternative leaching agent such as thiosulfate can serve to replace cyanide for precious metal leaching. Despite the many advantages of hydrometallurgical methods for producing precious metals, solid-phase extraction (SPE) has been widely used for the recovery of precious metals. SPE is suitable for metal recovery from low abundant sources such as industrial effluent or analytical detection of trace metal species. Among the various types of solid-phase sorbents available, activated carbon (AC) is a commonly used industrial sorbent for gold extraction. Activated carbon is a microporous amorphous carbonaceous material with varying levels of functional groups (e.g., carbonyl (C=O), hydroxyl (–OH), amino (–NH_2_), and thiol (–SH)) groups on its surface depending on the mode of preparation [2]. These functional groups may serve as electron donors (Lewis base) and may contribute to the formation of metal-pi interactions with metal cations (Lewis acid) on the AC surface [2]. In the case of inorganic adsorbents, clays are widely known, but they pose operational challenges as sorbents for recovery operations due to their fine particle size and colloidal properties in aqueous media [3]. Gold ions in the presence of suitable anionic chelator species such as cyanide or chloride in acidic media can form complexes with a resulting negative surface charge. In the case of AC as the sorbent for SPE, the uptake capacity of such gold anion complexes by AC may be reduced due to the negative surface charge of the graphene sites of AC and the occurrence of unfavorable interactions in such SPE systems. A comparison of AC-based materials with chitin-based biomaterials reveals several key differences, such as the density of polar functional groups, in accordance with the offset in the hydrophile-lipophile character of carbonaceous and biomaterial systems. In turn, the synthetic versatility and structural diversity of chitin-based sorbents have provided the motivation to examine their structure-adsorption properties in relation to conventional AC sorbents [4].

Chitosan is a copolymer comprised of β-(1,4)-glucopyranose units similar to cellulose; however, chitosan contains randomly distributed N-acetyl-glucosamine and glucosamine residues. The partial deacetylation of chitin biopolymers yields chitosan through enzymatic or chemical deacetylation methods, where chitin is second in abundance among natural polymers after cellulose. A primary biomass source of chitin is crustacean shells, such as those from crabs and shrimp; however, insects and fungi are also sources of chitin. Chitin and chitosan biopolymers differ in their structure according to the degree of acetylation at C-2. Chitin is fully acetylated, whereas chitosan contains either amine or N-acetyl groups (-NHR; where R = H and R = acetyl). As the level deacetylation of chitin reaches 50% or greater, the resulting biopolymer is more abundant with glucosamine units and is denoted as chitosan. Its solubility is greater than that of chitin in aqueous acidic media [5,6,7]. The degree of deacetylation (DDA) is an important structural feature that distinguishes chitin and chitosan, where potentiometric or spectral analysis via FTIR and NMR represent standard methods for estimating the DDA [7]. The amine groups of chitosan (Figure 1) are much more reactive than the acetamide groups of chitin. The modification of the amine groups underlies much research that aims at tailoring its *structure-function* properties for a range of applications [8]. For example, the uptake capacity of precious metals by pristine chitosan polymers is somewhat limited and may depend on the DDA, crystallinity, morphology, and molecular weight of the pristine biopolymer [7]. X-ray diffraction (XRD) was employed to estimate the crystallinity of chitosan, whereas viscometry results can provide estimates of the biopolymer molecular weight.

In the case of pristine chitin-based sorbents such as chitosan, the adsorption properties toward cation and anion species by the pristine biopolymer have limitations due to the low surface area and limited accessibility of surface chemical groups [8,9,10]. Chitosan can be modified by various physical and chemical synthetic strategies to yield biosorbents that yield environmentally friendly SPE materials with improved adsorption properties that are cost-effective for the recovery of precious metals from aqueous media [9]. Chitosan and its derivatives represent promising adsorbent materials for binding metals due to the presence of abundant heteroatoms and polar functional groups, which serve as potential metal-ion complexation sites [8,10,11]. The formation of cross-linked chitosan or grafted chitosan are two common methods for the synthetic modification of this versatile biomaterial. As well, the formation of chitosan-based composite sorbents as supports for metal or metal oxide nanoparticles (NPs) allow for further modification of the physicochemical properties of chitosan, which may afford greater uptake of metal-ion species. While there is a broad range of biopolymer adsorbent materials, chitosan and its modified forms have attracted great interest due to the relative abundance and synthetic versatility of this biopolymer platform. The opportunity to design tailored materials for specific adsorption-based applications that position chitosan as an emerging renewable material for controlled recovery of precious metal-ions in aqueous media.

This contribution provides a review of chitosan-based materials for their application as sorbents for solid-phase extraction (SPE) for precious metals, where the research results related to Au and Pd species covers literature over the past decade. The synthesis of chitosan derivatives with different heteroatom (N, O, and S) containing groups for cross-linked, grafted forms, and composites that contain metal oxides are described. As well, an overview of the adsorption properties at variable conditions for chitosan-based SPE materials with precious metals are outlined in the Tables presented herein. We aim to provide insight on the adsorption mechanism by the chitosan derivatives with various metal-ion systems, which will contribute to improved materials for the adsorption/desorption and the recovery of precious metal species. Chitosan-based adsorbents are viewed to contribute favourably in specialized applications that range from environmental remediation to mineral tailings recovery of precious metals [7].

## 2. Physical Modification of Chitosan

Pristine chitosan materials can be converted into different morphologies such as powders [12,13], nanoparticles [14], gel beads [15,16], membranes [17], sponges [18], and fibers [5,19] through physical and chemical methods. The pristine form of chitosan powders or flakes are not well suited as effective adsorbents for the uptake of precious metals because of the low porosity and surface area limitations due to the semi-crystalline nature of pristine chitosan. These features contribute to reduced surface accessibility of the active polar adsorption sites (e.g., -OH, NHR; R = acetyl or H) of chitosan. In turn, the accessibility of active functional groups for metal-ion binding affects the thermodynamic and kinetic properties of adsorption for pristine chitosan [20]. The slow diffusion rate reported for pristine chitosan indicates the low intra-particle mass transfer effect with metal ions and pristine chitosan, which results in a low uptake capacity of precious metals for this biopolymer. In general, the use of smaller chitosan particles, especially cross-linked forms, can result in a greater intra-particle mass transfer effect for such chitosan-based adsorbents. Mahaninia and Wilson reported on the use of bifunctional cross-linkers such as glutaraldehyde and epichlorohydrin for the cross-linking of chitosan to improve its overall stability toward dissolution in acidic media, along with an increase in the textural properties of the adsorbent (porosity and surface area) [21]. Spherical beads of cross-linked chitosan are preferred over powder materials, mainly for the ease of recovery over multiple adsorption-desorption cycles, especially for fixed-bed column applications. In particular, cross-linked chitosan beads showed relatively high uptake for precious metals when compared with covalently grafted chitosan sorbents, as described in the following section and by the examples presented in Table 1.

The alternative use of beads or pellets improves the mechanical stability and permeability of the resins, improves mass transfer, and prevents clogging in a fixed bed column through better control of the residence time in fixed-bed filtration systems [25]. However, chitosan fiber materials are not as good as chitosan beads for packing in a fixed bed system due to the irregular hydrodynamic behavior related to the fiber morphology [25]. By contrast, chitosan fibers with good mechanical strength are suitable for incorporation into membrane-based fiber composite filtration systems for harvesting precious metals.

## 3. Covalently Grafted Chitosan

### 3.1. N-Containing Chitosan Derivatives

Introducing functional units such as amino acids and aromatic amines with a linker or without a linker can result in functionalized chitosan composites. Depending on the types of N-containing functional groups in the grafted chitosan biopolymer, the number of the anion-exchange sites on the polymer will increase upon grafting with N-containing functional groups [26].

Azarova et al. showed that the N-(5-methyl-4-imidazolyl) methyl-grafted chitosan had excellent selectivity towards Au(III) and Pd(II) [26,27]. Pd(II) chloride forms square planar complexes with negative charges shown in Figure 2 [26]. The positively charged surface of N-(5-methyl-4-imidazolyl) methyl-grafted chitosan binds with the negatively charged Pd(II) chloride complexes. The 1 M thiourea/0.1 M HCl solution was used for the recovery of Au(III) and Pd(II), including the regeneration of the adsorbent after the adsorption process. Wang’s group developed 4′-nitro -4-aminoazobenzene-grafted chitosan to adsorb Au(III) and Pd(II). The optimum pH for extraction was at pH = 3 for Au(III) and Pd(II) ions at room temperature [28]. A 1 M thiourea/0.5 M HCl aqueous solution was suitable for the recovery of more than 80% of Au(III) from the grafted chitosan polymer over three adsorption/desorption cycles. Liang et al. reported a similar mechanism between precious metals and amidoxime-modified polyethylene fibers where elemental Au was found on the fibrous adsorbent surface after recovery of the metal. This result indicated that the mechanism of Au(III) recovery from an aqueous solution involved the reduction of Au(III) to Au(0) for the reaction between Au(III) and the –NH_2_ group of the amidoxime moiety [29]. In Figure 3, Liu et al. showed that electrostatic interactions and chelation both contribute to the adsorption mechanism as part of the uptake process for precious metals onto the chitosan composite material [30]. Table 2 provides a summary of some N-functionalized chitosan adsorbents for the removal of Au- and Pd-species from aqueous solution.

### 3.2. O-Containing Chitosan Derivatives

Park’s work showed that glutaraldehyde cross-linked chitosan beads could selectively separate Au(III) from other precious metals. In his work, the Au(III) ions were reduced to their elemental form (Au(0)) by the glutaraldehyde (GA) linkers. Greater GA content in the cross-linked chitosan beads resulted in better extraction efficiency for gold ions [23]. Similar results were also reported in Bui’s research, where Au(III) ions were extracted from an acidic e-waste leachate solution using glutaraldehyde cross-linked chitosan beads [22]. Table 3 provides a summary of O-containing covalently grafted chitosan for the removal of Au- and Pd-species from aqueous solution. The installation of crown ethers onto modified chitosan presents another material with high potential for SPE-based applications for the uptake of trace levels of metal ions [32,33]. Yang et al. developed 3-hydroxy-1, 5, 8-triaza-cyclodecan chitosan with high selectivity for Ag(I) ions [32]. The crown ether ligand has a strong Au(III) ion binding affinity, whereas crown ether-modified chitosan displays a relatively low sorption capacity towards Au ions. More extensive cross-linking and a greater hydrophobicity provide a lower degree of swelling for the crown ether-grafted chitosan [32]. The increased hydrophobicity impedes the accessibility of the metal ions to the crown ether binding sites. The crown ether-grafted chitosan adsorbent is suitable for increasing the selectivity towards specific metal ion systems, where an example is illustrated in Figure 4.

Pang et al. grafted pyromellitic dianhydride onto the amine groups of chitosan to install carboxylic groups that bind with Pd(II) ions from an aqueous solution [34]. Their results obtained using extended X-ray absorption fine structure (EXAFS) spectroscopy reveal that two chloride ions in the outer shell of Pd were replaced by the O atoms of the carboxylate groups on the phenyl moiety, as shown in Figure 5. The grafted carboxyl ligands can undergo ligand exchange for chloride ions upon binding to Pd(II) ions [34].

**Table 3 molecules-27-00978-t003:** O-containing covalently grafted chitosan for the removal of Au- and Pd-species from aqueous solution.

Adsorbent	Characterization Methods	Metal Ions	Uptake Q_m_ (mg/g)	Conditions				Literature Refs.
				pH	T (K)	Time (h)	Dosage (g/L)	
GMCCR	FTIR, SEM, BET	Au(III), Pt(IV), Pd(II)	169 (Au), 122 (Pt), 120 (Pd)	1–4	298	24	3.33	[35]
PAA-CS	FTIR	Gold cyanide	10.18	7	298	24	1	[36]
LMCCR	FTIR, BET, SEM	Au(III), Pt(IV), Pd(II)	70 (Au), 109 (Pd), 129 (Pt)	2	303, 313, 323	0.1–7	0.01–0.4	[37]

CS: cross-linked chitosan; GMCCR: glycine-modified cross-linked chitosan resin; LMCCR: l-lysine-modified cross-linked chitosan resin; PAA: poly acrylic acid.

### 3.3. S-Containing Chitosan Derivatives

Another strategy to increase the selectivity of chitosan biopolymers towards precious metals is to inhibit the influence of metal speciation by the preparation of grafted chitosan polymers that are less susceptible to the presence of competitor ion binding effects. To address this goal, the preparation of a thiolate-modified chitosan led to a change in the sorption mechanism of the ion exchange resin. The introduction of a new metal chelator (thiourea) binding sites led to greater selectivity for the adsorption of Au(III) species. Table 4 provides a summary of S-containing covalently grafted chitosan for the removal of Au- and Pd-species from aqueous solution. Arguelles–Monal and Peniche–Coves used thiourea and epichlorohydrin to modify chitosan, where the FTIR spectra showed the C=S group stretching region (1050–1300 cm^−1^) is attached to the amide/amine region of the chitosan, resulting in changes to the C-N stretching at (1590–1420 cm^−1^) [38]. Once the available amide/amine groups from the chitosan were fully reacted upon grafting with the thiourea groups, whereas the addition of extra linkers such as glutaraldehyde did not undergo effective bonding with the amide/amine groups of chitosan. The FTIR spectra for the C=O bond observed near 1710 cm^−1^ for glutaraldehyde cross-linked sorbents did not change much after grafting with the additional thiourea groups [39]. Monier’s research showed that the thiol-modified chitosan could have a maximum Au(III) uptake capacity of 370 mg/g [40]. The chitosan composite was prepared by grafting 2-mercaptobenzaldehyde, which revealed that favourable interactions occur with Au(III) ions preferentially. An Au(III) imprinted cross-linked biopolymer composite was synthesized by cross-linking chitosan with epichlorohydrin and leaching the Au(III) template ions out of the final product [40]. Zhao made cystamine-modified magnetic chitosan adsorbents that revealed a maximum uptake capacity of 478.5 mg/g for Au(III). These composite adsorbents incorporate nanoscale magnetic particles, which enable the facile recovery of the adsorbents under a magnetic field to afford recycling over multiple adsorption cycles [41].

Figure 6 shows that a thiophene-chitosan hydrogel was prepared by reacting chitosan with thiophene-2-carboxaldehyde in an ethanol environment to yield the resulting composite. The thiophene-chitosan (TCS) chitosan in Figure 6 showed good uptake towards soft acid atoms such as Hg(II).

**Figure 6 molecules-27-00978-f006:**
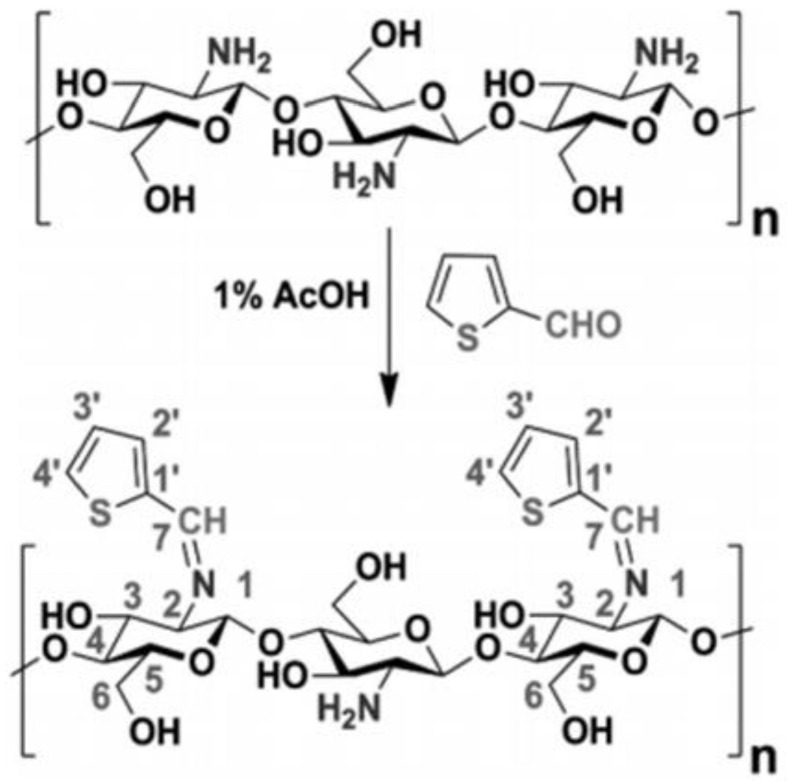
Synthesis of a thiophene-chitosan (TCS) hydrogel. Reprinted with permission from [42].

**Table 4 molecules-27-00978-t004:** S-containing covalently grafted chitosan for the removal of Au and Pd species from aqueous solution.

Adsorbent	Characterization Methods	Metal Ions	Uptake Q_m_ (mg/g)	Conditions				Literature Refs.
				pH	T (K)	Time (h)	Dosage (g/L)	
CDF-CS	FTIR, SEM, XPS, and Zeta-potential	Au(III)	478	1–10	298, 308, 318	6	0.4	[41]
ETSA	FTIR, XPS, XRD, SEM	Au(III)	6	1–3	303, 313, 323	25	0.2	[43]
Thiourea GCC	SEM,	Au(III)	5.0	1–4	298, 308, 318, 328	12	1.00	[44]
TSC-CCB	FTIR, SEM, XPS, Zeta-potential	Au(III)	1470	6	298	3–24	0.4	[45]
Thiocarbamoyl Chitosan	XPS, ^13^C NMR, FTIR, Elemental analysis	Au(III), Pt(IV), Pd(II)	0.8(Pt), 2.1(Pd), 2.3(Au)	2	298	18	0.050	[46]
CS-MoS_2_	SEM, EDS, FTIR, XRD, XPS, and Zeta-potential	Au(III)	3108	1–9	303, 314, 323	6	0.4	[47]
CS-GTU	SEM, EDS, XRD, XPS	Au(III)	741	1–9	303, 313, 323	30	0.4	[48]

CS: cross-linked chitosan; TSC-CCB: thiosemicarbazide functionalized corn bract; CDF-CS: cystamine-modified magnetic cross-linked chitosan; ETSA: thiourea-grafted alginate; CS-GTU: guanylthiourea formaldehyde cross-linked chitosan.

### 3.4. Other Chitosan and Biopolymer Derivatives

Graphite and chitosan-based composites have been reported as sorbents for solid-phase extraction of heavy metal ions [49]. Dai et al. developed multi-wall carbon nanotube systems grafted with chitosan to adsorb V(V), Cr(VI), Pd(II), and As(V), where 0.5 M HNO_3_ was used to elute adsorbed metal ions from this graphite-based modified chitosan for ICP-MS analysis. The high enrichment factors 52–128 of these metals onto the solid adsorbent led to relatively low detection limits to 1.3–3.8 ng/L (ppb) using this SPE-based method [49]. In Table 5, activated carbon (AC) had a higher binding capacity than chitosan with Au(III) due to the larger surface area and more accessible binding sites of AC. In the gold cyanidation process, the gold-bearing solution has a value of pH > 10. The binding mechanism between AC and gold metal ions could be due to cation-pi interactions. The activated carbon-modified chitosan revealed Au(III) species were adsorbed at pH 8 but with a much lower adsorption capacity value. The chitosan-modified AC has a higher pH_pzc_ value (7.5) versus typical ACs with a lower pH_pzc_ value (6.5). For adsorption of Au(S_2_O_3_)_2_^3−^ ions, the uptake of gold was reduced as the pH of the gold solution increased above 7.5. The negatively charged surface of the chitosan-modified AC adsorbent repels the Au(S_2_O_3_)_2_^3−^ ions.

Incorporating inorganic silica oxide species within a chitosan hydrogel structure was shown to increase the surface area of the hydrogel by creating separation among the chitosan-based fibers into more layers. In turn, this led to an increase in the space available within the chitosan fiber network for metal species to diffuse to the active sites. He et al. developed chitosan/silica gel adsorbents for the selective pre-concentration of Cd(II) and Cu(II), where the enrichment factor was estimated as 166.7, and the detection limits for these metals were 20 and 38 ng/L, respectively [50]. Ngatijo used quaternary amines modified with silica to remove Au(III) from a gold mining effluent. The Au(III) sorption capacity was 74.5 mg/g at pH 5 [51].

Recently, Gao made a thiourea-modified alginate for the simultaneous removal of Au(III), Pd(II), Pt(II), Co(II), Ni(II), and Cu(II) ions under acidic conditions. The thiourea-modified alginate composite (cf. Figure 7 for the molecular structure) was found to selectively bind with Au(III) ions through coordination and electrostatic interaction from hydroxyl, carboxyl, pronated amino, and –C=S groups [43].

**Table 5 molecules-27-00978-t005:** Equilibrium adsorption capacities of chitosan, activated carbon, and modified activated carbon materials with Au(III) species under variable experimental conditions in aqueous solution.

Adsorbent	Characterization Methods	Metal Ions	Uptake Q_m_ (mg/g)	Conditions				Literature Refs.
				pH	T (K)	Time (h)	Dosage (g/L)	
Chitosan	FTIR	Au(III)	33	2	298	0.5	1.0	[31]
Chitosan	FTIR, XPS	Pd(II)	n/a	0.5–6	298	0.5	0.2	[52]
Activated carbon (AC)	SEM, PSD	Au(III)	67	1–11	298–348	6	0.03–0.18	[53]
AC	SEM-EDS, FTIR, XPS	Au(I)	25.9	6–10	308, 318, 328	3–50	0.05	[54]
Modified AC	pH_pzc_, SEM, XPS, BET	Au(III)	3.55	8.5	298, 308, 318	25	1.00	[55]

Gao achieved an Au(III) uptake capacity of about 184.8 mg/g at 323 K with epichlorohydrin/thiourea-modified porous alginate adsorbents, which is an outstanding result for a recyclable polysaccharide-based adsorbent [43]. Polyethylenimine (PEI)-grafted alginate fibers showed a pronounced Au(III) uptake capacity at 2300 mg/g [56]. The high uptake of gold ions was related to electrostatic interactions and a reduction reaction by the amines, hydroxyl, and aldehyde groups of the PEI-grafted alginate fibers [56]. Guo’s research involved the use of carboxyl-functionalized diethylaminoethyl cellulose, where a high Au(III) uptake capacity was achieved [57]. S-containing covalently grafted biosorbents showed good selectivity towards Au(III) among N-, O-, and S-containing covalently grafted biosorbents because the soft S-ligand tends to bind with soft Au(III) ions better than other N- and O-containing ligands. The total Au(III) uptake capacity for S-, N-, or O-containing covalently grafted chitosan might be lower when compared to cross-linked chitosan-based sorbents. This relates to cross-linked chitosan sorbents due to an increased surface area to interact with metal ions. However, the protonated amine groups on the cross-linked chitosan are bound with the metal ions through electrostatic interactions that yield limited selectivity toward Au(III) ions. Nonetheless, cross-linked chitosan and grafted chitosan sorbents showed greater uptake of Au(III) versus AC and its modified sorbents. AC contains negatively charged graphenic surface sites, which reduces the interaction with anionic species such as AuCl_4_^−^ ions.

Although the functionalization of chitosan can increase the number of binding sites and binding affinity towards precious metal ions, it is also important to find optimal equilibrium and kinetic conditions (temperature, pH, dosage, and extraction time) for the adsorption process.

## 4. Conditions Influencing Batch Adsorption of Precious Metals

Various conditions are known to influence the efficiency of the adsorption properties of adsorbent-metal ion systems, as follows: (*i*) temperature; (*ii*) pH effects; (*iii*) adsorption capacity of the adsorbent; (*iv*) initial metal-ion concentration; (*v*) presence of competitor ions; and (*vi*) mixing and contact time. The role of these parameters on the equilibrium uptake properties in batch mode is further outlined in the following sections.

### 4.1. Temperature

The temperature conditions for the adsorption process are a key factor affecting the uptake kinetics of the chitosan-based adsorbents because of the key role of temperature on the diffusion rate of precious metal species in aqueous solutions along with the temperature-dependent swelling properties of chitosan [28]. Similarly, the temperature will also affect the intraparticle diffusion of the adsorbate on the adsorbent surface, along with the mass transfer of the metal species from the bulk solution phase to the binding sites at the adsorbent surface sites or within the micropore domains [28,34]. Grad and coworkers developed dibenzo-18-crown-6-ether-modified chitosan sorbents to remove palladium and platinum species. These sorbents can adsorb more Pd(II) ions with increasing temperature [58]. On the other hand, the thermodynamic adsorption parameters are also dependent on temperature [59], where the uptake capacity increases with increasing temperature for such endothermic processes. The sign of the enthalpy of the adsorption process provides insight into the role of temperature effects for the uptake process. The standard enthalpy change for an adsorption process was obtained with the use of a van ’t Hoff analysis of the temperature dependence of the adsorption isotherms, along with the Gibbs free energy relationship. Positive enthalpy indicates an endothermic sorption process, and negative enthalpy indicates an exothermic sorption process, according to the slope of the van ’t Hoff plot. The standard Gibbs free energy of the reaction is described by Equation (1). A sorption process is spontaneous when it yields a negative difference in the standard change for the Gibbs free energy for the overall process.
ΔG^°^ = ΔH^°^ − TΔS^°^(1)
where G is the Gibbs free energy and H and S are the enthalpy and the entropy of the system, respectively. The Gibbs free energy isotherm equations are described by Equation (2) through the relationship between ΔG^°^ and the equilibrium constant (*K_eq_*) for the adsorption process, according to the constant obtained by the best-fit value obtained from a Langmuir isotherm model.

Equation (2) reveals that the van ’t Hoff relation can be used to determine the enthalpy and entropy information of the sorption process. By performing sorption experiments at different temperatures, the enthalpy (*H*) and entropy (*S*) of the sorption process are obtained from the linear regression parameters (slope, intercept) by plotting a graph of *lnK_eq_* versus the inverse temperature (−1/T).
(2)$$lnK_{eq}=-\frac{\Delta H{}^{\circ}}{RT}+\frac{\Delta S{}^{\circ}}{R}$$

Wang et al. showed that the uptake of Au(III) ions using a thiourea-grafted chitosan composite revealed an exothermic adsorption process [28]. When the Au(III) adsorption isotherms were obtained at variable temperature conditions between 25 to 35 °C, the uptake capacity of Au(III) ions by the thiourea-grafted chitosan was reduced by 50% at elevated temperatures.

### 4.2. Solution pH

The solution pH is an essential factor in the precious metal adsorption process because the solution pH will result in protonation or deprotonation of the active sites of the adsorbent and affect its surface charge, along with possible changes in the metal speciation in aqueous solution [59,60]. Gold ions in the presence of excess chloride are normally present in solution as an anionic complex in the form of tetrachloroaurate species, especially at low pH in the presence of chloride ions. In alkaline media, colloidal precipitates of Au(OH)_3_ can form as the pH is raised above 4. Hence, many studies have focused on adsorption conditions at acidic pH from pH 1.0–4.0. Wang et al. found that thiourea-grafted chitosan had a point of zero charge (pH_pzc_) of 4.5. For pH < 4, the surface charge of the thiourea-grafted chitosan remained positive, which favors the binding of anionic gold species. The optimal pH for Au(III) ions uptake was near pH 4 for the thiourea-grafted chitosan [44]. Similar trends in the effect of pH for Au(III) adsorption were reported by An et al. [61], which concur with results reported elsewhere, where the optimum pH for the uptake AuCl_4_^−^ was between pH 2–3 [61,62]. At low pH, gold is present as an anionic chloride complex, favoring binding with positively charged adsorbent surfaces by electrostatic attraction. In particular, chitosan-based adsorbents with protonated amine groups are active sites for the adsorption of chloride-based anion gold complexes [35,63,64]. As long as the pH of the solution does not result in deprotonation of the chitosan composite, the overall surface charge of the composite remains positive [63]. Conventional mining processes used by industry employ the cyanidation process to extract gold as anion complexes from the ores into an aqueous solution. Swantomo’s work showed that chitosan-polyacrylamide graft copolymers have a maximum uptake towards gold cyanide at pH 7 [36]. Some adsorbents such as pyromellitic anhydride modified with ultrahigh molecular weight polyethylene chelates Pd(II) ions with a maximum uptake at pH 6 [59]. Zhao’s work also indicated that greater Au(III) uptake occurs near pH 6 and becomes attenuated for pH > 7 [41].

### 4.3. Adsorption Capacity and the Number of Binding Sites of Adsorbents

Since the maximum adsorption capacity of the adsorbents can be estimated based on an analysis of the adsorption isotherms at equilibrium, the number and availability of active adsorbent sites can affect the level of adsorbate removed. When the initial precious metal concentration is constant, a large adsorbent dosage in the adsorption process may lead to a lower uptake capacity since the number of binding sites available exceeds the metal-ion species. As a result, the active sites of the adsorbent are not fully saturated. Usually, the adsorption efficiency increases with the adsorbent dosage and reaches a plateau upon saturation of the adsorption sites for the case of monolayer adsorption (Langmuir adsorption isotherm Equation (3)).
(3)$$Q_{e}=\frac{Q_{m}K_{L}C_{e}}{\left(1+K_{L}C_{e}\right)}$$

*Q_m_* (mg/g) is the maximum amount of the adsorbate bound onto the monolayer of the adsorbent, *Q_e_* (mg/g) is the amount of the adsorbate bound at equilibrium, *C_e_* (mg/L) is the unbound adsorbate concentration in solution at equilibrium, and *K_L_* is the Langmuir adsorption constant. In contrast, the adsorption capacity of the adsorbent per unit mass decreased after the point when the adsorption efficiency reached the maximum value [41,65]. Huiping et al. showed that the gold sorption capacity per unit mass of the bacteria decreased after the maximum adsorption yield at a dosage of 10.0 g/L [66]. Therefore, the lower uptake capacity at higher dosage values was due to the presence of excess active sites on the adsorbent at conditions that exceed a dosage of 10 g/L for such equilibrium conditions.

### 4.4. Initial Source Concentration of Precious Metals

The initial ion concentration can increase the adsorbent uptake capacity by shifting the equilibrium toward the bound state for a given adsorbent dosage. In the equilibrium adsorption study, when the amount of the adsorbent is fixed, the number of active sites on the adsorbent surface for binding is constant compared to the number of metal ions in the aqueous solution. Tofan’s work showed that higher Au(III) uptake was achieved at an initial concentration of 114 mg/L, as compared with 57 mg/L [60]. Fujiwara et al. studied the effect of the initial concentration of Au(III) ions from 20 to 200 mg/L by fixing the L-lysine modified cross-linked chitosan resin dosage at 3.33 g/L at pH 1.0 [37]. The experimental results showed an increased level of adsorbed gold ions by the modified chitosan when the initial concentration of Au(III) ions increased, but the level (%) of the Au(III) ions adsorbed decreased as the initial concentration of Au(III) ions are increased [37].
(4)$$\Delta G{}^{\circ}=-RT\left(lnK_{eq}\right)$$
(5)$$K_{eq}=\frac{k_{f}}{k_{r}}$$

In Equation (5), *k_f_* is the forward reaction rate constant, *k_r_* is the reverse reaction rate constant, and *K_eq_* is the equilibrium constant in the limit where the concentration of reactants and products approach a steady-state condition. The higher initial concentration of sorbate yields a value of *K_eq_* > 1, which results in a negative change in the standard Gibbs energy according to Equation (4) for the sorption process. A spontaneous sorption process may give a high amount of sorbate uptake, but the adsorption efficiency of the sorbents was reduced upon increasing the initial sorbate concentration. Zhao and coworkers showed similar results, where the amount of Au(III) adsorbed increased from 300 to 475 mg/g as the initial Au(III) concentration increased from 200 to 900 mg/L [41].

### 4.5. Competing Ions in the Precious Metal Matrix

For the adsorption of gold species by the electrostatic interaction of functional groups on chitosan, the sorption capacity of the precious metals is highly influenced by other competitor anion species present in the solution. Various trends were noted for the ion affinity with the ion-exchange resins based on their physicochemical properties: (1) higher valence ions tend to bind with greater affinity onto chitosan polymers, (2) greater polarizability of the ions binds better with sorbents, (3) ions with larger ionic charge/ionic radius ratios have higher binding affinity to sorbents. For instance, the PtCl_6_^2−^ species bond with chitosan more strongly than Cl^−^ because of the ionic charge/ionic radius ratio order Cl^−^ (5.5 × 10^3^ pm^−1^) < PtCl_6_^2−^ (6.4 × 10^3^ pm^−1^) < SO_4_^2−^ (8.7 × 10^3^ pm^−1^) [63]. For platinum adsorption by glutaraldehyde cross-linked chitosan composites, co-adsorption of sulfate ions have more effect on the total uptake capacity of the composite than chloride ions [63], in line with observations reported by Steiger et al. for a chitosan-alginate ternary complex that contained Al(III). The uptake of Pt(IV) in hydrochloric acid solutions is ca. six-fold greater than Pt(IV) in sulfuric acid solution. This trend indicates that the binding affinity of sulfate ions by the amine groups on the chitosan composites exceeds that for singly charged ions, especially those that are singly-charged that have a larger size and delocalization effects. Steiger et al. also found sulfate binds strongly to the amine groups of chitosan [10]. The FTIR spectra of C=N bonds around 1666 cm^−1^ were well resolved for the thiourea-grafted chitosan in an aqueous HCl environment, but these C=N bonds were broadened at 1590 cm^−1^ for the thiourea-grafted chitosan in sulfuric acid solution [39]. In acidic pH conditions, gold, platinum, and palladium species are present as anionic complexes, which are bound to positively charged protonated amine groups on the surface of chitosan. However, the uptake of platinum and palladium decreased in the presence of chloride and nitrate ions due to competition for positive sites on the surface of the chitosan composite [63]. Therefore, the sorption performance of modified chitosan is not solely based on the global concentration of competing ions present in the solution. As well, sorption depends on the binding affinity of these ions with the active sites of the modified chitosan composites.

In addition, the influence of ionic strength and electrolyte type (NaCl and NaNO_3_) on the uptake of Pt(IV), Au(III), and Pd(IV) was reported by Ramesh et al. [35] They proposed an indirect method to model the inner- and outer-sphere coordination of precious metal complexes, which are affected by the presence of additional chloride ions. The results indicated that the inner-sphere coordination of precious metal complexes was not overly affected by an increase in ionic strength. However, the outer-sphere coordination of these complexes was largely affected by the presence of additional NaCl [35]. In other words, the competitor ions have a negligible effect on the chelation-based adsorption mechanism versus the electrostatic adsorption mechanism. Regarding the alkaline and alkaline-earth metals, pristine chitosan biopolymers have no affinity to these metal species in an acidic environment, and the modified chitosan with soft Lewis base ligands will not undergo bonding with these alkaline-earth metals either [7]. Liu’s results were compared for the uptake of modified chitosan with Au(III), Pd(II), Cu(II), Ni(II), Cr(III), and Al(III), where it was found that the chelation effect of Au(III) cations was the most potent among the tested metal-ions [30]. Lin’s research used the hard-soft acid-base (HSAB) theory to account for the trend observed for the S-modified chitosan adsorbents, where the greatest selectivity was noted for Au(III) amongst a mixture of competitor cation species: Co(II), Cd(II), Ni(II), and Pd(II) [45].

### 4.6. Mixing Rate and Contact Time

There are several steps in the solid-liquid heterogeneous adsorption process. The first step involves the transport of the solute (sorbate) from the bulk liquid phase to the surface of the sorbent. Then, intraparticle diffusion occurs at specific sites within the sorbent particle. The second step involves the adsorption mechanism itself. Since the first step involves external diffusion and intraparticle diffusion, providing an effective mixing rate can reduce the role of diffusion effects in the external diffusion process for the transfer of solutes onto surface sites and within the pore domains of the solid phase adsorbent. With an effective agitation rate, the kinetics of the adsorption process depends on the intraparticle diffusion onto the solid adsorbents and adsorbate binding with the active sites of the adsorbent. In physical adsorption, the time needed for adsorbate binding with active sites on the adsorbent is almost instantaneous. The main rate-determining step would be the intraparticle diffusion step in the physical adsorption process. In the case of chemisorption, the chemical reaction rate between adsorbates with the adsorbent should be accounted for in the overall adsorption process. Therefore, both the intraparticle diffusion and the chemical reaction rate steps may interfere in determining the rate-limiting steps for adsorption that involves chemisorption [11]. Overall, the agitation rate and contact time between adsorbate and adsorbent are essential factors in determining the kinetic rate constants of adsorption. It is also necessary to consider the diffusion equations, the sorption isotherm equation, and the kinetics of the reaction for complete modeling of the overall adsorption kinetics. However, the equation for complete modeling the sorption kinetics is more complex, where a simplified approach considers the diffusion steps that control the kinetic rate-determining steps [15,41]. Alternatively, the pseudo-second-order (PSO) kinetic model has been used to describe the solution-solid adsorption process. Experimental estimates of the adsorption kinetics can be used to describe how much time is needed to saturate the adsorbents with bound adsorbate [41,67]. The pseudo-first-order (PFO) kinetics model can be used to describe sorption kinetics [68], where the PFO model is shown in Equation (6):(6)$$q_{t}=q_{e}\left(1-e^{-k_{1}t}\right)$$
where *q_e_* and *q_t_* are the amounts of adsorbate bound to the adsorbent (mg/g) at a steady state and variable time (*t*; min). The pseudo-second-order (PSO) rate constant *k*_2_ is described by Equation (7).
(7)$$q_{t}=\;\frac{k_{2}q_{e}^{2}t}{\left(1+k_{2}q_{e}t\right)}$$

The PSO rate constant (*k*_2_) is determined from a non-linear best-fit to the experimental data, where *q_t_* is the level of adsorbate uptake at a variable time (*t*) by the adsorbent. Depending on the adsorbent type, the adsorption capacity of the adsorbate increases rapidly within the first 20 min, and 90% of active sites of the adsorbent reach saturation.

## 5. Mechanism of Adsorption

As outlined in Table 1, Table 2, Table 3, Table 4 and Table 5, various groups have studied the mechanism of adsorption between chitosan biopolymers with precious metal species. Generally, chemical and physical adsorption are essential aspects of the adsorption process between precious metal ions and the functional groups of chitosan-based adsorbents. Physical adsorption mechanisms include electrostatic interactions such as hydrogen bonding, dipole-dipole interactions, and π-π interactions. Chemisorption includes complexation, chelation, and redox reactions to yield precipitates [30]. The adsorption mechanism between Au(III) ions and chitosan-based sorbents are viewed as a complex process that involves electrostatic interactions, redox reactions, and other processes. On the other hand, the adsorption of Au(III) ions by AC is driven mainly by cation-π interactions with the graphene surface sites.

### 5.1. Physical Sorption Mechanism

Glutaraldehyde cross-linked chitosan composites were studied to evaluate the uptake of anionic gold species such as the tetrachloroaurate anion in acidic media. Cross-linking serves to impart stability for chitosan in acidic media since dissolution can be reduced due to the presence of cross-links at the amine sites, limiting the solvent accessibility. The formation of imine linkages also presents enhanced Coulombic attraction due to the efficient protonation of the adsorbent up to alkaline pH values, attributed to the greater pK_a_ of imine groups of the cross-linked adsorbent. The anionic gold complex species can undergo adsorption by a range of electrostatic interactions due to various active sites, where the gold uptake capacity of the chitosan composite is reduced upon deprotonation of the imine groups above pH 11. This is in accordance with the higher pK_a_ of imine versus than for amine groups [30,64]. Pethkar et al. used X-ray absorption near-edge spectroscopy (XANES) and extended X-ray absorption fine structure spectroscopy (EXAFS) to support that Au(III) ions can interact with the protonated carbonyl and carboxyl groups by a physical sorption mechanism [69]. Xue and Wilson have proposed the role of the –OH and –NHR (R = H or –COCH_3_) groups of chitosan participate in the formation of complexes with Cu(II) [13,70], shown in Figure 8. The relatively high binding affinity was confirmed due to the negligible levels of leaching in chitosan-based Cu(II) composites by Hassan et al. [70]

### 5.2. Chemisorption Mechanism

For the case of chemisorption, the process of Au(III) uptake by modified chitosan was monitored by XRD, where the presence of reduced Au(0) was evidenced on the surface of the saccharide units. The presence of aldehyde groups was reported to donate electrons that result in a reduction of Au(III) to Au(0), where the formation of micro precipitates of gold nanoparticles (NPs) on the surface of modified chitosan sorbents was observed by electron spin resonance (ESR) spectral results. Romero-Gonzalez et al. provided experimental support for these Au NPs, which were assigned to various morphological forms based on ESR results. The various morphology of the Au NPs are as follows: hexagonal platelets, tetrahedral, rods, and decahedral Au NPs. The Au(III) sorption properties of chemically modified chitosan were studied by Donia et al., where the primary interaction between modified chitosan polymers and Au(III) involves the formation of salt linkages between R-CH=NH^+^ and AuCl^4−^ species [71]. In a report by Sun and Song, they indicated that the catalytic reduction of Au(S_2_O_3_)_2_^3−^ occurs on the surface sites of a MoS_2_@Fe_3_O_4_ composite [72]. These composites can produce photoelectrons in the conduction band that creates holes on the valence band to reduce adsorbed Au(S_2_O_3_)_2_^3−^ to Au(0) directly [72]. Zhao used energy-dispersive X-ray spectroscopy (EDS) to determine that gold ions were adsorbed onto the surface of cystamine-modified magnetic cross-linked chitosan. XPS analysis showed N1*s*, and S2*p* spectral results indicate that the –NH and S groups combined with AuCl_4_^−^ by chelation and by ion-exchange processes. The Au4*f* spectrum showed that the Au(III) was reduced to Au(I) and Au(0) [41]. Liu’s results report that the N1*s* spectral band shifted to higher energies, and C-NH_3_^+^ must interact with Au(III) [30]. Similarly, Lin et al. [45] reported an adsorption mechanism, where the lone pair electrons of the N-atoms for an amino group and the O-atoms of the carboxyl group of the modified chitosan serve as the active sites of the adsorbent. These heteroatom groups can undergo chelation and subsequently reduce the Au (III) species [45]. These results were confirmed in other studies using FTIR spectral analysis of C-O and C-N bond stretching of the adsorbent after the adsorption process [37]. In general, these lone pair electrons are available through various heteroatoms (O, N, and S) from the hydroxyl, amino, and thiol groups, which can supply electrons to the vacant atomic orbitals of Au(III). The chemical bonding for this mechanism is the dominant process for the capture of Au(III) in acidic environments with O-, N- and S-modified chitosan composites. In the XPS results reported by Zhao et al., the lone pair of electrons on the N-atom of the poly(itaconic acid) grafted chitosan were donated to the metal species. The decreased electron density around the N-atom and the increase of the binding energy in the XPS N 1*s* spectra after Au(III) adsorption indicated the formation of a covalent bond between modified chitosan and Au(III) [47,48]. The chelation properties of chitosan follow the rules of HSAB theory, as defined by Pearson [73], where many S-containing ligands on the chitosan polymer are considered soft Lewis bases. These soft Lewis base sites can donate lone electron pairs to those soft Lewis acids such as precious metals to form coordinate covalent bonds [45].

## 6. Desorption and Recovery of Precious Metals

There are various factors to consider in the selection of the desorption solution for the stripping (washing off) of precious metal species from the chitosan composites after the adsorption process, as follows: (1) the effective removal of precious metals; (2) minimizing mechanical damage to the chitosan composites; (3) environmentally friendly or less hazardous chemical (stripping) agents; and (4) cost-effectiveness [74]. Ichikawa et al. found that a 0.1 M NaOH solution can wash off 95% adsorbed Au(III) ions, and 0.1 M NaCN can wash off 92% adsorbed Au(III) from an eggshell membrane adsorbent [75]. Ramesh et al. found that 0.7 M thiourea with 2 M HCl solution effectively washes off the precious metals from a glycine-modified cross-linked chitosan composite, where the composite did not show any decreased uptake of precious metals after five adsorption-desorption cycles [35]. Donia et al. suggested the use of a 0.5 M acidified thiourea solution to strip Au(III) from chemically modified chitosan, where the modified chitosan showed no reduced performance after four cycles [76]. The results reported by Wang et al. showed that 0.5 M thiourea and 2 M HCl could have a 99% desorption efficiency to remove Au(III) from the modified chitosan sorbents [44].

## 7. Strategies for Enhancing the Adsorption Capacity

### 7.1. Pretreatment to Optimize Conditions of the Process

In the case of palladium adsorption processes, sulfur ions are soft bases that have a tendency to bind with soft acids such as gold ions. The elimination of competitor ions, such as sulfate, will increase the uptake capacity towards Pd-species. In the case of a physical adsorption process, the electrostatic interaction is the dominant process, where shifting the precious metal speciation to the favored forms to bind at the active sites of the modified chitosan composites will result in greater uptake capacity of the target metal ion species. By controlling the pH of the solution, the desired form of metal species will predominate, and the surface charge of the modified chitosan can be optimized to yield an increase in the uptake capacity.

### 7.2. Increasing the Active Sites of the Chitosan Composites

For the case of chemisorption or the physisorption processes, an increase of accessible coordination sites between the precious metals and the modified chitosan composites will result in an increased adsorption capacity. In the case of the chelating precious metals with active functional groups on the modified chitosan composite, the precious metal uptake capacity increases as the extent of grafting of N- and S-based groups increase onto the chitosan polymer backbone [63].

### 7.3. Overview of the Solid-Liquid Extraction of Gold Ions by AC and Modified Chitosan Sorbents

The conventional method of gold extraction is through the cyanidation process (Figure 9A). Anion complexes such as [Au(CN)_2_]^−^ ions in alkaline solution are produced in the cyanidation process. In general, gold ions interact with sorbents through electrostatic interaction. These negatively charged ions prefer to bind with sorbents that have positively charged surfaces. For the gold cyanidation process, the alkaline condition makes a negatively charged surface of activated carbon. These AC-based sorbents compete with cyanide ions to bind with Au(I) ions in alkaline solutions. To reduce the interference from cyanide ions, sorbents with a positively charged surface are preferred that can interact with such gold anion species. Most modified chitosan sorbents with pH_pzc_ around 8.5 could show positively charged surfaces in the range of pH 1 to 8.5. Replacing the gold cyanidation process with a thiosulfate process (Figure 9B) can produce Au(S_2_O_3_)_2_^3−^ ions to bind with modified chitosan sorbents more efficiently.

The conceptual flow charts show different leaching methods chosen for gold extraction produce variably charged gold complexes. The binding mechanism between the sorbent and the gold species should be considered, along with the gold leaching methods to select the proper adsorption media for extracting gold to meet the efficiency and sustainability requirements of the process. While activated carbon is a popular adsorbent material for the recovery of precious metals in hydrometallurgical processes, there is a need to explore other types of adsorbents since AC does not favourably adsorb the gold thiosulfate complex, whereas ion exchange resins are susceptible to resin poisoning, resin swelling, co-adsorption, and passivation of the adsorbent surface. The potential utility of modified chitosan sorbents is revealed by their unique uptake properties toward precious metals at the laboratory scale. Many reports document the study of powdered adsorbent materials, whereas few studies have described the limitations related to the use of powdered adsorbents (such as the effects of high back pressure) in fixed bed columns. Further studies are needed to characterize powdered materials and the other sorbent morphologies to enable large-scale applications under variable conditions relevant to industrial processing and for environmental remediation [77,78]. The practical utility of other types of adsorbents with morphologies such as beads, granules, and pelletized systems for fixed-bed column studies is recommended for implementation in dynamic separations in flow systems for pilot studies and eventual scale-up [77]. Pilot-scale studies are recommended to evaluate the techno-economic aspects of biopolymer and activated carbon adsorbents for the separation and recovery processes illustrated in Figure 9 for gold species over multiple cycles. In the case of hydrometallurgy, the sustainable processing of precious metals will require further innovation in metal leaching and recovery (cf. Figure 13 in [79]), as outlined in a critical review by Hsu et al. [79].

## 8. Conclusions

A summary of various chitosan derivatives for the uptake of precious metals was provided in this contribution, according to various synthetic strategies summarized in Table 1, Table 2, Table 3 and Table 4. In particular, the synthesis and characterization of cross-linked chitosan and chitosan grafted with functional groups that contain N-, O-, and S-modified ligands are described. The adsorption properties of the modified adsorbents were compared against the properties of pristine chitosan and activated carbon (Table 5). Modified forms of chitosan display markedly variable uptake efficiency toward precious metals, according to the use of variable experimental parameters at equilibrium, along with kinetic uptake conditions. In Section 4, variable pH presents unique conditions that distinguish chitosan over AC-based adsorbents, since many modified chitosan sorbents have positively charged surfaces between pH 1 to 8.5. Chitosan adsorbents can readily interact with negatively charged gold anion species by controlling the adsorbent surface charge through pH or via synthetic modification. The adsorption mechanism with precious metals was discussed, where electrostatic and redox reactions are the key factors that contribute to the adsorption mechanisms between modified chitosan sorbents and gold ions. The uptake of Au(III) ions by activated carbon is driven by cation-π interactions. Various strategies to increase the uptake capacity of precious metals by the modified chitosan composites were outlined and further reveal the potential scope for development of chitosan-based SPE maerials. With reference to the use of industrial adsorbents such as activated carbon for gold extraction, a relatively low gold uptake occurs in an acidic environment. By contrast, modified chitosan composites have favorable gold uptake between pH 2–4. In the case of metal recovery in aqueous media, strategies for the desorption and recovery of Au(III) suggest potential advantages for the application of modified chitosan to harvest precious metals. A future perspective for such biosorbents relates to tailoring the structure and properties of chitosan for selective uptake of target metal species in a mixed system. This goal is especially relevant to sustainable resource extraction and industrial recycling of metals from complex wastewater matrices. The development of such green adsorbent technology has considerable potential for industrial processing and recovery of precious metals, along with environmental protection in aquatic systems.

## Figures and Tables

**Figure 1 molecules-27-00978-f001:**
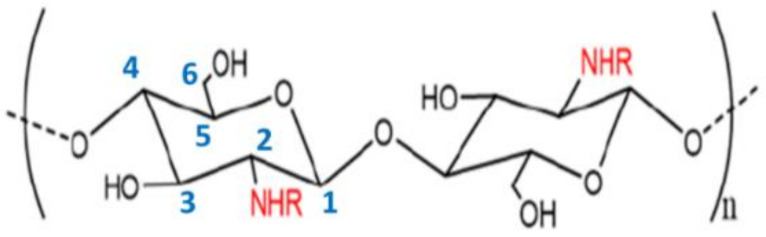
The molecular structure of chitin and chitosan according to the degree of acetylation (DAC) at C-2, where R=H or R=COCH_3_ for chitosan (DAC < 50%) and R=COCH_3_ for chitin (DAC = 100%). The degree of polymerization is denoted by n, and the carbon numbering assignment is denoted in blue font.

**Figure 2 molecules-27-00978-f002:**
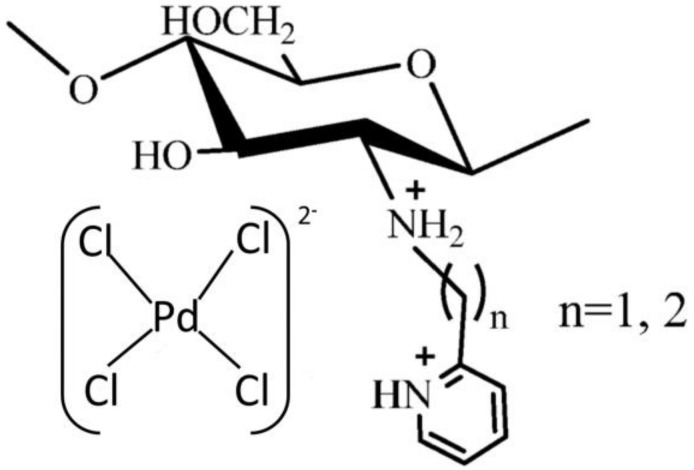
Proposed structure of Pd(II) anions chelation with a grafted ligand form of chitosan. Adapted with permission from [26].

**Figure 3 molecules-27-00978-f003:**
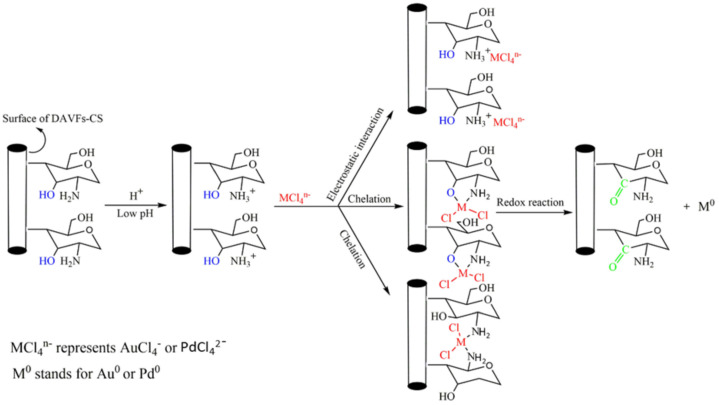
Proposed precious metal removal mechanisms on chitosan composites. Reprinted with permission from [30].

**Figure 4 molecules-27-00978-f004:**
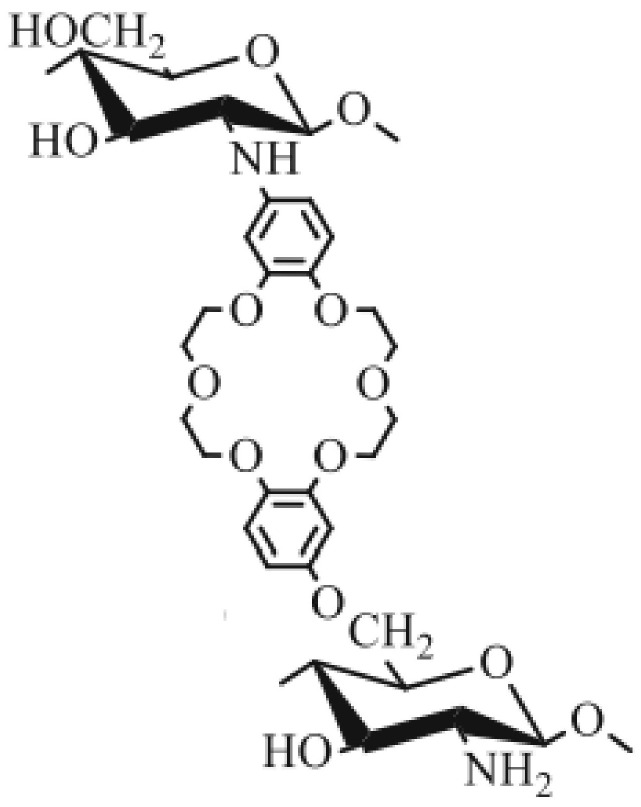
Crown ether-grafted chitosan, where the crown ether moiety was inferred to serve as an active site for metal ion binding. Reprinted with permission from [33].

**Figure 5 molecules-27-00978-f005:**
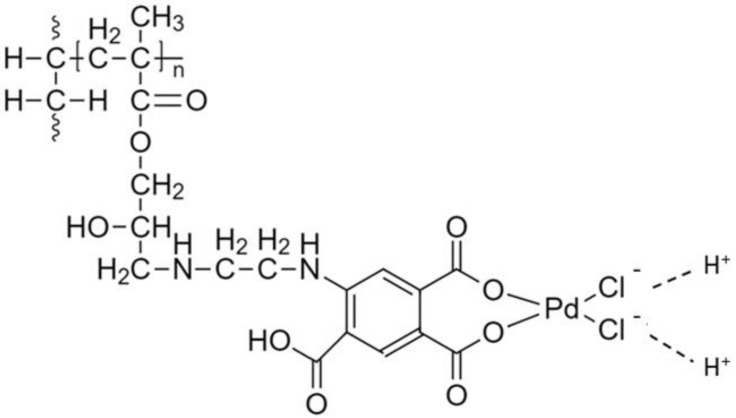
Mechanism of Pd(II) adsorption onto the surface of the UHMWPE-PMDA fibers. Adapted with permission from [33,34].

**Figure 7 molecules-27-00978-f007:**
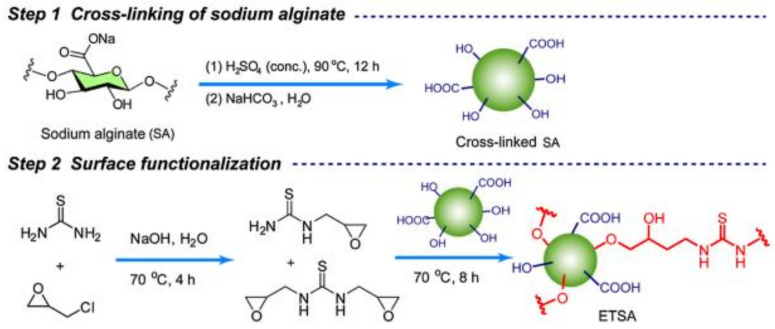
Thiourea-grafted alginate bead system. Reprinted with permission from [43].

**Figure 8 molecules-27-00978-f008:**
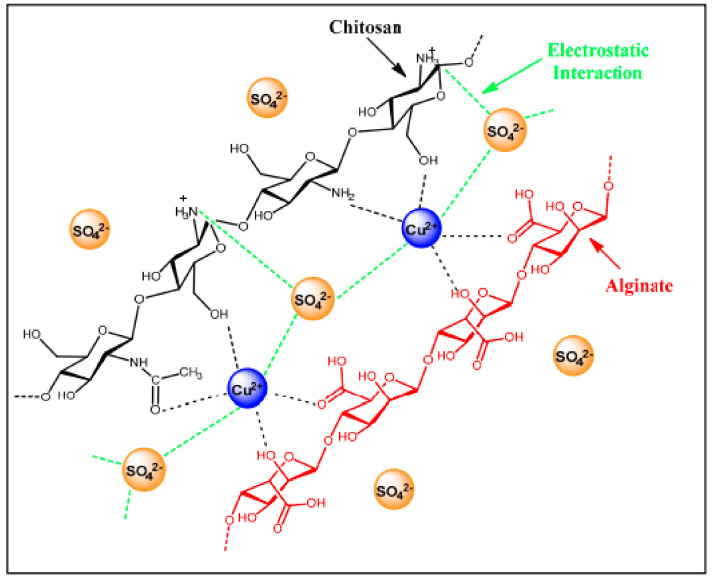
Proposed adsorption mechanism for SO_4_^2−^ anions by a ternary biopolymer composite that contains Cu(II). Reprinted with permission from [70].

**Figure 9 molecules-27-00978-f009:**
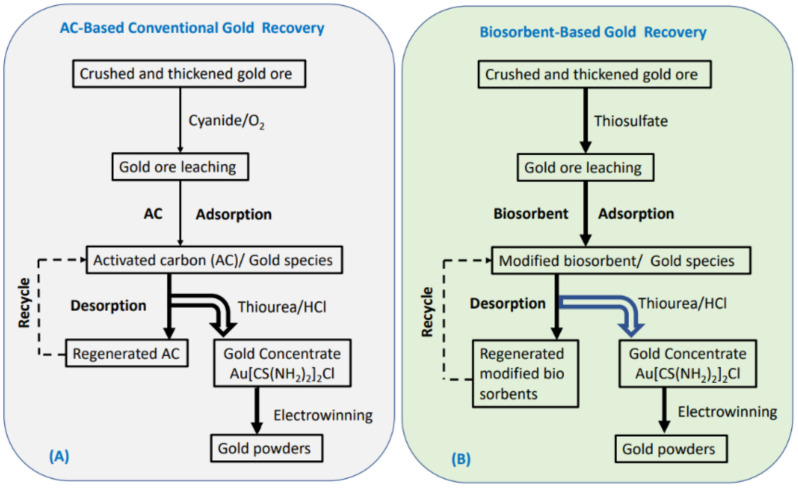
Conceptual illustration for the gold extraction process by different adsorbents: (**A**) AC-based conventional gold recovery, and (**B**) biosorbent-based gold recovery.

**Table 1 molecules-27-00978-t001:** Cross-linked chitosan for the removal of Au- and Pd-species from aqueous solution.

Adsorbent	Characterization Methods	Metal Ions	Uptake Q_m_ (mg/g)	Conditions				Literature Refs.
				pH	T (K)	Time (h)	Dosage (g/L)	
GCC beads	SEM, XPS, TGA, FTIR	Au(III)	600	2	283, 298	24	0.2–1.5	[22]
GCC Beads	XPS, TGA, FTIR	Au(III), Pd(II)	240 (Au), 120 (Pd)	1–10	298	0.5 to 24	1–15	[23]
GCC beads	TGA, SEM, EDX,	Au(III)	400	0.2–4	298	24	1	[24]

GCC: glutaraldehyde cross-linked chitosan; Q_m_: the maximum amount of the adsorbate bound onto the monolayer of the adsorbent was estimated from Equation (3).

**Table 2 molecules-27-00978-t002:** N-functionalized chitosan adsorbents for the removal of Au and Pd species from aqueous solution.

Adsorbent	Characterization Methods	Metal Ions	Uptake Q_m_ (mg/g)	Conditions				Literature Refs.
				pH	T (K)	Time (h)	Dosage (g/L)	
IMC	XPS, AAS, ^1^H NMR, & XRD	Au(III), Pt(IV), Pd(II)	158 46 68	2–9	298	18	1.00	[27]
DAVF-CS	XPS, ^13^C NMR, FTIR, TGA, and elemental analysis	Au(III), Pd(II)	322 (Au), 207 (Pd)	1–6	298	24	0.2	[30]
NCMC	FTIR	Au(III)	30	2–12	298	30	2	[31]
		Au(III), Pd(II)	69.9 (Au), 58.6 (Pd)	2, 3				[28]

CS: cross-linked chitosan; IMC: N-(5-methyl-4-imidazolyl) methyl chitosan; NCMC: N-carboxymethyl chitosan; DAVF: dialdehyde viscose fibers. Q_m_: the maximum amount of the adsorbate bound as a monolayer onto the adsorbent was estimated by use of Equation (3).

## Data Availability

Not applicable.

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
