# Peer review of "An Overview of Modified Chitosan Adsorbents for the Removal of Precious Metals Species from Aqueous Media"

_molecules, 2022, doi:10.3390/molecules27030978_

Round 1

Reviewer 1 Report

In general, the review manuscript is well organized and quite smoothly written with clarity, except some minor improvement as suggested below:

Title should reflect the overall of the manuscript. The title of the manuscript is too general, since the review mostly cover chitosan, I would suggest to change “biopolymer” to “chitosan”.

Is all the chitosan discussed in the manuscript nanostructured? In title, it stated “nanostructured biopolymer” but in the discussion (section 2.0 till the end), author did not mention or discuss about the size of the chitosan in all the reported studies. For example, section 3.2, line 200, Park’s work….chitosan beads…it is very common beads are in micron size, however, it not stated here the size of the beads. So I suggest either to remove “nanostructured” from the title or need to improve the discussion part.

Qm in Table 2 was calculated using Equation 3 same as Table 1? Please add as footnote in Table 2.

Figure 3, 5 are blurred. Please replace the Figure with clearer image.

Page 8, line 259, “TCS” should be put the full name for the first time stated in the text.

Page 11, line 338, “Pang and Wu’s research 338 showed glycidyl methacrylate and pyromellitic dianhydride grafted polyethylene fibers 339 can adsorb more Pd(II) ions with increasing temperature 34.” Can author try to find other example that used chitosan or other biopolymer in this discussion?

Page 14, line 477-479, “where the greatest selectivity was noted for Au(III) amongst a mixture of competitor cation species: Co(II), Cd(II), Ni(II), Pd(II), and “Au(III)”? Au(III) competitor cation to Au(III)?

Author Response

Author Response to the Reviewer Comments on MS ID: molecules-1561341

Reviewer #1

In general, the review manuscript is well organized and quite smoothly written with clarity, except some minor improvement as suggested below:

Title should reflect the overall of the manuscript. The title of the manuscript is too general, since the review mostly cover chitosan, I would suggest to change “biopolymer” to “chitosan”.

Is all the chitosan discussed in the manuscript nanostructured? In title, it stated “nanostructured biopolymer” but in the discussion (section 2.0 till the end), author did not mention or discuss about the size of the chitosan in all the reported studies. For example, section 3.2, line 200, Park’s work….chitosan beads…it is very common beads are in micron size, however, it not stated here the size of the beads. So I suggest either to remove “nanostructured” from the title or need to improve the discussion part.

Response:  We agree with the review’s comments. To address the comments, we have changed the title as follows: “An Overview of Modified Chitosan Adsorbents for the Removal of Precious Metals Species from Aqueous Media”

Qm in Table 2 was calculated using Equation 3 same as Table 1? Please add as footnote in Table 2.

Response:   The footnote for eqn was added as recommended by the reviewer

Figure 3, 5 are blurred. Please replace the Figure with clearer image.

Response: A high-resolution version of Figure 3 was added, and a new version of Figure 5 was added (see markup version of the manuscript).

Page 8, line 259, “TCS” should be put the full name for the first time stated in the text.

Response:  The full name was added for TCS, as recommended.

Page 11, line 338, “Pang and Wu’s research 338 showed glycidyl methacrylate and pyromellitic dianhydride grafted polyethylene fibers 339 can adsorb more Pd(II) ions with increasing temperature 34.” Can author try to find other example that used chitosan or other biopolymer in this discussion?

Response:  Another reference describing a modified chitosan with dibenzo-18-crown-6-ether substituents was added to replace the example noted by the reviewer.

Page 14, line 477-479, “where the greatest selectivity was noted for Au(III) amongst a mixture of competitor cation species: Co(II), Cd(II), Ni(II), Pd(II), and “Au(III)”? Au(III) competitor cation to Au(III)?

Response:  An appropriate correction was made to address the typographical error. We have deleted Au(III).

In summary, the authors wish to acknowledge Reviewer #1 for the insightful and constructive comments, along with the opportunity to improve the overall quality of this paper. The paper was further edited for language, syntax, and clarity throughout to meet the high standards of this journal.

Reviewer 2 Report

This work presents an overview of nanostructured biopolymer adsorbents for the removal of precious metal species from aqueous media. The theme is interesting and is within the scope of the journal, however, the manuscript needs to undergo some changes. Following are my detailed suggestions for future improvements, and then it can be accepted.
abstract graphic
Please build an abstract figure.

Abstract
Authors must include in the abstract the main results and conclusions obtained from the works that were reviewed, such as the maximum adsorption capacity of adsorbents.
Authors should improve the resolutions in figures 3 and 5.
It is not common to place figures in the introduction, I suggest that authors remove figure 1 and place it together in the construction of the abstract figure.
Authors should build a topic related to future perspectives in this area for adsorption. In this aspect, the gaps that need to be filled can be mentioned, such as studies in continuous systems, application of statistical physics models, neural networks, economic analysis of the developed adsorbents aiming at large-scale application and competitive adsorption containing more than one adsorbate.

Author Response

Reviewer #2

This work presents an overview of nanostructured biopolymer adsorbents for the removal of precious metal species from aqueous media. The theme is interesting and is within the scope of the journal, however, the manuscript needs to undergo some changes. Following are my detailed suggestions for future improvements, and then it can be accepted.
abstract graphic
Please build an abstract figure.

Response: A revised version of the TOC graphic was prepared as recommended.

Abstract
Authors must include in the abstract the main results and conclusions obtained from the works that were reviewed, such as the maximum adsorption capacity of adsorbents.

Response: The abstract was revised to address the reviewer comment (see markup version) by the addition of key highlights in the revised version.

Authors should improve the resolutions in figures 3 and 5.

Response: Figures 3 and 5 were revised accordingly to address the reviewer comment. A high-resolution version of Figure 3 was prepared and Figure 5 was redrawn.

It is not common to place figures in the introduction, I suggest that authors remove figure 1 and place it together in the construction of the abstract figure.

Response: While we agree that the inclusion of figures are not commonplace in research papers, the current contribution relates to a review of modified chitosan. To provide essential background on chitosan and chitin in this introductory section, it seems appropriate to retain Figure 1 since it provides a context for the role of structure-function referred within the introductory section and throughout the following sections that follow.

Authors should build a topic related to future perspectives in this area for adsorption. In this aspect, the gaps that need to be filled can be mentioned, such as studies in continuous systems, application of statistical physics models, neural networks, economic analysis of the developed adsorbents aiming at large-scale application and competitive adsorption containing more than one adsorbate.

Response: Some recommendations for future work and perspectives that serve to advance the field were provided near the end of section 7.3. Refer to the markup version of the manuscript.

In summary, the authors wish to acknowledge Reviewer #2 for the insightful and constructive comments, along with the opportunity to improve the overall quality of this paper. The paper was further edited for language, syntax, and clarity throughout to meet the high standards of this journal.

Round 2

Reviewer 2 Report

The authors made all the corrections and changes requested. Therefore the article can be accepted for publication.